# Dispositional and ideological factor correlate of conspiracy thinking and beliefs

**Jan Ketil Arnulf**[1], **Charlotte Robinson**[2], **Adrian Furnham**[1]*

**1** Department of Leadership and Organisational Behaviour, Norwegian Business School (BI), Nydalsveien, Oslo, Norway, **2** Department of Psychology, University of Bath, Bath, England

* adrian@adrianfurnham.com

**Data Availability Statement:** All relevant data are within the paper and its Supporting Information files.

**Funding:** The author(s) received no specific funding for this work.

## Abstract

This study explored how the Big Five personality traits, as well as measures of personality disorders, are related to two different measures of conspiracy theories (CTs)The two measures correlated r = .58 and were applied to examine generalisability of findings. We also measured participants (N = 397) general knowledge levels and ideology in the form of religious and political beliefs. Results show that the Big Five and ideology are related to CTs but these relationships are generally wiped out by the stronger effects of the personality disorder scales. Two personality disorder clusters (A and B) were significant correlates of both CT measures, in both cases accounting for similar amounts of variance (20%). The personality disorders most predictive of conspiracy theories were related to the A cluster, characterized by schizotypal symptoms such as oddities of thinking and loose associations. These findings were corroborated by an additional analysis using Latent Semantic Analysis (LSA). LSA demonstrated that the items measuring schizotypal and related symptoms are cognitively related to both our measures of CTs. The implications for the studying of CTs is discussed, and limitations are acknowledged.

## Introduction

Conspiracy theories (CTs) are essentially the belief that the causes of most major social, political and economic events are due to a plot by multiple, evil, people with a selfish, political goal in mind [1]. It has been argued, but disputed, that they usually form part of a *monological* belief system [2] in the sense that people have a *conspiracist worldview*, or thinking style. This means that they accept and integrate new CTs on a wide range of issues, which can be reliably measured [3–5]. It has also been suggested that people accept often strange and outlandish CTs because they serve a psychological function for people who feel powerless, excluded or disadvantaged [1,2,6,7].

In many ways CTs could be seen as superstitious, magical, and paranormal beliefs which has attracted people, and consequent research interest, over the years [8,9]. The question posed is why people believe in CTs when there is no credible scientific evidence for them; that is what functions do they fulfil? [10]. One way of examining this issue is to examine individual difference correlates of CTs. It should be noted that belief in CTs and conspiracy thinking are used essentially synonymously as the latter is seen to lead to the former.

**Competing interests:** The authors have declared
that no competing interests exist.

This study attempts to move the literature on CTs forward in three ways. First, we use two different CT questionnaires to check the generalisability of results. There are a number of different measures of general CTs as well as measures of very specific beliefs about conspiracies in science, business, politics etc [5]. Further, the study is a part replication of Furnham and Grover [11] but using additional individual difference variables. They used the 15 item measure devised by the Swami research group working in the area [4]. Second, we examine personality disorder (PDs) correlates of CT beliefs which has hitherto been ignored. There have been a number of studies examining the DarkTriad and CTs [12–16] but these examined only two of the many PDs recognised by recent editions of American and European psychiatric manuals [17]. Indeed, there are good theoretical reasons to believe that the other disorders (i.e. schizotypy) are more strongly related to CTs [18]. For example, belief in unlikely conspiracies might have more in common with schizotypal thought disorders [19] and paranoia than with the calculative cold-heartedness of anti-social typologies [20]. Third, we examine the association between general knowledge and the belief in CTs.

There are long lists of topics that the CT researchers have investigated, including medicine [21] commerce [22], and very specific events [23]. These keep constantly being "updated" as in the case of COVID-19 [24]. However, because it is assumed that people tend toward "conspiracist outlooks and world views", various researchers have developed questionnaires which ask participants to what extent they accept *a range* of relatively well-known theories. The fact that they nearly all demonstrate high internal reliability (as measured by Cronbach's alpha) is taken as indication of the propensity of some individuals to accept and endorse a wide range of theories and integrate them into their belief system. In this study, we examine two measures of CTs, using them to see if we can generalise our findings [2,4].

Over the last decade there has been dramatic rise in studies in this area by people from many disciplines including political science, psychology, and sociology [25–29]. Many studies have examined individual difference correlates of CTs including ability, ideology, and personality [30–32]. The COVID-19 pandemic has been a great impetus to research in this area [33–36]. One difficulty in reviewing and comparing them is that they have used different measures trying to assess belief in CTs.

Recently, in a study of over 11,500 people from East and West Europe and America Walter and Drochon [2] tested and confirmed number of hypotheses, using a 10-item generic scale of CTs used in this study. They tested a number of hypotheses: First, the *education hypothesis* namely higher the education level, the lower people's propensity to conspiracy thinking. Second, the *right-wing hypothesis*: that the more people perceive themselves as positioned towards the right of the political spectrum, the higher their propensity to conspiracy thinking. Third, the *ideological extremity hypothesis*: which stated that people who perceive themselves as positioned on the ideological extremes of the political spectrum have a higher propensity to conspiracy thinking than respondents who perceive themselves in the centre of the political spectrum. Fourth, the *magical thinking hypothesis*: people who have a high level of magical thinking have a greater tendency to conspiracy thinking which is associated with people who feel powerless and excluded. Fifth, the *representation within the political system hypothesis*: people who do not feel represented within the political system are more inclined to conspiracy thinking than people who do feel represented within the political system. Sixth, the *institutional trust hypothesis*: people who have low levels of institutional trust are more inclined to conspiracy thinking than people who have high levels of institutional trust. Seventh, the *economic insecurity hypothesis*: people who feel insecure about their economic situation have a higher propensity to conspiracy thinking. They confirmed most of their findings noting that CT believers were more inclined to position themselves on the right of the political spectrum, engage in magical thinking, feel distrust towards public officials and reject the political system.

The relationship between demographic (sex, age, education), ideological factors (religion, politics), and conspiracist beliefs has been explored [32]. Those who endorse CTs misattribute a great deal of agency and intentionality to others where it is clearly inappropriate to do so [27]. Many suggest that education is the best way to reduce belief in CTs [37]. It has also been demonstrated that religious and superstitious people are more likely to endorse CTs [38,39]. As regards personality trait correlates Goreis and Voracek [40] in a meta-analysis of studies that used personality theories concluded that that neither trait Agreeableness, nor Openness-to-Experience, or the other personality factors were significantly associated with conspiracy beliefs if effect sizes were aggregated. More recent studies however suggest that these relationship warrant revisiting [41].

In a study highly relevant to this Furnham and Grover [18] asked 475 adults to complete measures of Belief in CTs [4], two personality disorders tests as well as a short intelligence test and two self-evaluations. Belief in CTs was correlated with nearly all PDs as well as the three established higher order clusters (A- odd and eccentric; B- dramatic, emotional or erratic disorders, C- anxious or fearful disorders). A series of hierarchical regressions showed five of the variables were significant which indicated that less intelligent participants, scoring higher on two PD clusters (Cluster A and B) but lower on Cluster C believed more in the CTs. The present study was a part confirmation of Furnham and Grover [18] with four exceptions: we used two CT measures; we used a different, more robust, measure of IQ; we also measured non-clinical personality variables; and we examined ideological correlates (political and religious) beliefs. We aimed to examine both clinical and non-clinical traits in the same study.

In this study we examine four sets of factors that have been hypothesised and shown to be related to CTs. Our primary aim is the attempt to replicate the findings on the same population but using different CTs measures, as well as determine the amount of variance explained by each set of factors. We chose three classic demographic factors: sex, age and education and hypothesised the latter would be significantly associated with belief in CTs. That is higher education would be negatively correlated with belief in CTs (H1). We also used a measure of intelligence to test the hypothesis that higher IQ test scores would be negatively correlated with belief in CTs (H2). Secondly, we measured ideology with items asking about political and religious beliefs. We attempted to replicate previous findings to test the hypothesis that more religious people would endorse CTs (H3) and more politically conservative/right-wing people would endorse CTs (H4).

Our third set of variables was personality. Studies that have examined trait correlates of CTs have shown mixed results, but where correlations are significant they are modest. We examine two hypotheses based on the previous literature: that trait Agreeableness will be negatively (H5) but Openness positively (H6) related to CTs.

Our fourth set of variables was the personality disorders (PDs). People with Cluster A personality disorders tend to experience major disruptions in relationships because their behavior may be perceived as peculiar, suspicious, or detached. People who have a personality disorder from Cluster B tend to either experience very intense emotions or engage in extremely impulsive, theatrical, promiscuous, or law-breaking behaviors. People with personality disorders in Cluster C tend to experience pervasive anxiety and/or fearfulness. We predict a positive correlation between endorsing CTs and Cluster A (H7) and Cluster B (H8).

Finally, we wanted to explore how the cognition involved in conspiracy theories is related to the clinical dimensions of personality. The clusters B and C of personality disorders are primarily related to the individual's ability to regulate emotions in the relationships to other people. Cluster A however involves cognitive components of egocentric or magical thinking with odd or loose associations related to the cognitive peculiarities of psychosis in general and schizophrenia in particular. This type of thinking has shown itself to be detectable through

digital text analysis [19,42,43], more specifically Latent Semantic Analysis (LSA). This technique has recently gained more interest as a method in psychology [44–46] as it allows a statistical comparison of cognitive content in texts, adding to the use of rating scales [47,48]. The thought patterns peculiar to schizophrenia and Cluster A conditions have been found to show up in statistically significant ways using LSA, predicting these conditions better than human judges [49]. For this reason, we will subject the item contents to LSA to shed light on the response patterns [50], an approach akin to what has earlier been found to work the diagnostics of personality and psychopathology in the schizotypal spectrum [19,47,49]. If cognitive peculiarities are indeed central to a propensity for believing in CTs we expect that the semantics of the conspiracy theories are most closely related to the semantics of the items measuring schizotypal PDs (H9). The technique itself will be explained in more detail below.

## Method

### Participants

In all 397 people took part in this study: 195 male, 199 female and 3 non-binary, They ranged in age from 19 to 71 with a mean of 39.9 years (*SD* = 11.63 yrs). In all 54% were graduates; 93% were British nationals, and 60.3% owned their own homes. They were all working and indicated their occupation which were very varied to include accountants, health workers and people in IT. Asked their company rank, 5.0% indicated they were the CEO, 4.2% directors, 22.2% managers and 68.7% employed. They also rated their beliefs on various 10 point scales: Religious (Not at all = 0 to Very = 10) 2.29 (SD = 2.90); Politics (Conservative = 0 to Liberal = 10) 5.55 (SD = 2.46).

### Measures

*Conspiracy Thinking* [2], which henceforth will be referred to as "CT1". This was a 10-item scale devised as part of the Conspiracy and Democracy project at the University of Cambridge. It consisted of 10 statements that are generic in nature and not connected to any specific societal, economic or political systems. The scale was administered to over 11,000 people and was examined for its psychometric properties. In this study the Alpha was .60, which is lower than the usual .70 cut-off for acceptability.

*Belief in Conspiracy Theories* (BCTI), which will henceforth be referred to as "CT2" [4]. This is a 15-item measure that describes a range of internationally-popular conspiracy theories. Participants rated their belief that each conspiracy was true on a 9-point scale, ranging from 1 (*Completely false*) to 9 (*Completely true*). An overall score was computed as the mean of all items, with higher scores reflecting greater belief in conspiracy theories. Scores on this measure have been shown to be one-dimensional [4] and correlate strongly with scores from a generic measure of conspiracist ideation (*r* = .88) [26]. In the present study, Cronbach's α for the BCTI was .91.

*The Mini-IPIP* [51] This is a 20-item short form of the 50-item International Personality Item Pool—Five-Factor Model measure. The Mini-IPIP scales, with four items per Big Five trait, had consistent and acceptable internal consistencies across five studies. The scales showed a comparable pattern of convergent, discriminant, and criterion-related validity (Studies 2–5) with other Big Five measures. In this study the reliabilities were Extraversion (.85) Neuroticism (.76) Openness (.74) Agreeableness (.78) and Conscientiousness (.67)

*General Knowledge Test* [52] is an open-answer item questionnaire [53]. The test has been used in numerous studies mainly done by Lynn and his colleagues. Scores were computed by adding together all correct answers (1 = correct; 0 = incorrect). We decided to use a short version which comprised 10 items like "*Who wrote 1984; What disease stops blood clotting? Which*

*Italian designer was shot in Miami in 1997? In what game can you bid a grand slam? Which is the principal street for finance in New York?*".

*Coolidge Axis-II Inventory–Short Form (SCATI)* [54]. This 70-item self-report measure assesses 14 personality disorders, 10 from *DSM-V*, 2 from Cluster B of the *DSM-IV-TR* (Depressive and Passive Aggressive) and 2 from *DSM-III-R* (Sadistic and Self-Defeating). The SCATI has good internal scale and test-retest reliability [55]. It has been used to predict PDs in subclinical [56] and clinical [57] populations. Using the DSM-5 classification the three clusters were calculated: A (odd and eccentric, alpha = .73), B (dramatic, emotional or erratic disorders) (alpha = .72), C (anxious or fearful disorders, alpha = .73).

## Procedure

Latent Semantic Analysis (LSA) is a prevalently used vector-based representation of meaning in language that can be used for computational purposes [58,59]. The general principle behind the technology is to sample texts from naturally occurring sources such as newspapers, books and other streams of text, generating a word database with hundreds of thousands of words occurring in prevalent contexts. These text samples are turned into semantic spaces through singular value decomposition (SVD), a technique akin to principal component analysis (PCA). Through this technique, it is possible to obtain stable numerical estimates on the likelihood of sequences of words used in the diagnosis of thought disorders [49] or how closely groups of words such as item texts or generated free texts assemble each other [60–62], for example how close a given text comes to a diagnostically recognized statement [47,63]. The numerical output of LSA is usually a cosine, where numbers approaching 1 indicate identical meaning between the compared texts, and lower numbers indicate disparate or unrelated meanings in the compared texts. Open implementations of LSA can be found in the statistical packages R [64] or Python [65], and as a simple interface on the open website www.lsa.colorado.edu.

Ethics permission was sought and received (CEHP/514/2017). Participants were recruited through Prolific.ac, an online participant database. Prolific was chosen over alternative online recruitment websites, due to its greater diversity of participants. We specified that people had to be employed. The survey took an average of 14 minutes to complete and participants were paid £2.00 after completing the survey. The usual inspection of the data was done at the end to look for irregularities (patterned results, excessive missing data, very short or long completion time) and very few were found. We have a standard procedure used in all studies.

## Results

We first analysed results from the 10 item Conspiracy Thinking measure. Table 1 shows the correlational results. Seven correlations with the CT measure were significant indicating that higher scores on CT were associated with non-degree and less intelligent, more politically conservative, Disagreeable and Conscientious as well as being higher on Cluster A (with odd and eccentric beliefs) and Cluster B (being dramatic, emotional and erratic). Whilst the correlation with Conscientious is significant but modest it does appear at odds with the previous literature.

We then performed a multiple hierarchical regression with CT and the criterion variable and four sets of variables as predictors: demography (sex, age, education and intelligence), ideology (religious and political beliefs); the Big Five traits and then the three PD clusters. Table 2 shows that four variables were significant indicating that higher CT scores were associated with not having a university degree, being more Conscientious, while having higher Cluster A and lower Cluster C scores. These accounted for just under a quarter of the variance.

**Table 1. Means, SDs and correlations between CT1, demographics, ideology, five-factor personality and the three clinical clusters.**

| | Mean/ Sum | SD | CT1 | Sex | Age | Ed.years | GK | Relig. | Pol. | Agree. | Consc. | Extrav. | Open. | Neurot. | Cl A | Cl B |
|---|---|---|---|---|---|---|---|---|---|---|---|---|---|---|---|---|
| CT1 | 1.25 | 1.24 | | | | | | | | | | | | | | |
| Sex | 1.51 | 0.50 | -.06 | | | | | | | | | | | | | |
| Age | 39.88 | 11.62 | -.07 | -.02 | | | | | | | | | | | | |
| Ed.years | 1.46 | 0.50 | .03 | -.03 | -.05 | | | | | | | | | | | |
| GK | 6.95 | 2.43 | -.12* | -.07 | .16*** | .02 | | | | | | | | | | |
| Relig. | 2.29 | 2.90 | .07 | .08 | .13* | -.06 | -.04 | | | | | | | | | |
| Pol. | 5.61 | 2.42 | -.21*** | .03 | -.15** | .17** | .10 | -.17** | | | | | | | | |
| Agree. | 15.58 | 2.61 | -.12* | .33*** | .10* | .04 | .06 | .14** | .18** | | | | | | | |
| Consc. | 13.42 | 2.95 | .10* | .00 | .00 | -.01 | .06 | .09 | -.12* | .04 | | | | | | |
| Extrav. | 11.08 | 3.43 | .04 | .03 | .05 | -.03 | .09 | .15** | -.02 | .25*** | .10* | | | | | |
| Open. | 14.45 | 2.94 | -.06 | -.08 | -.05 | .08 | .12* | -.06 | .23*** | .20*** | .07 | .17*** | | | | |
| Neurotic. | 11.95 | 3.29 | .06 | .26*** | -.23*** | .15** | -.15** | -.01 | .12* | .08 | -.31*** | -.19*** | -.03 | | | |
| Cl A | 27.59 | 7.00 | .31*** | .00 | -.19*** | .04 | -.11* | .03 | -.06 | -.19*** | -.20*** | -.29*** | -.03 | .47*** | | |
| Cl B | 34.75 | 7.41 | .20*** | -.05 | -.27*** | .08 | .02 | .04 | .04 | -.08 | -.26*** | .15** | .12* | .43*** | .54*** | |
| Cl C | 30.26 | 6.70 | .06 | .08 | -.24*** | .11* | -.04 | -.06 | .14** | -.09 | -.23*** | -.39*** | -.07 | .54*** | .73*** | .52*** |

Scores for the traits and PDs are summed; Sex: Male = 1, female = 2

$*p < .05$

$**p < .01$

$***p < 001.$

Last, we did the same hierarchical regression but using all 14 PDs rather than clusters. We recognize the obvious problem of multicollinearity, as correlations between these different measures are relatively high, as to be expected [54] The result was significant (F(25,332) = 6.21,

**Table 2. Regression results for CT1.**

| | B | Std. Error | Std. Beta | t |
|---|---|---|---|---|
| Sex | -.102 | .129 | -.042 | -.787 |
| Age | -.002 | .005 | -.025 | -.471 |
| School years | -.040 | .014 | -.145 | -2.877** |
| regious | -.000 | .021 | -.000 | -.004 |
| Politics | -.040 | .026 | -.084 | -1.538 |
| GK | -.031 | .028 | -.056 | -1.099 |
| Agreeableness | -.022 | .027 | -.047 | -.803 |
| Conscientiousness | .046 | .022 | .112 | 2.120* |
| Extraversion | .027 | .022 | .078 | 1.222 |
| Openness | -.023 | .022 | -.057 | -1.066 |
| Neuroticism | .020 | .023 | .056 | .877 |
| Cluster A | .078 | .013 | .454 | 5.615*** |
| Cluster B | .017 | .012 | .109 | 1.485 |
| Cluster C | -.058 | .016 | -.322 | -3.629*** |

$*p < .05$

$**p < .01$

$***p < 001.$

**Table 3. Means, SDs and correlations between CT2, demographics, ideology, five-factor personality and the three clinical clusters.**

| | Mean | SD | CT2 | Sex | Age | Ed.years | GK | Relig. | Politics | Agreeable. | Conscient. | Extrav. | Open. | Neurot. | Cl A | Cl B |
|---|---|---|---|---|---|---|---|---|---|---|---|---|---|---|---|---|
| CT2 | 42.43 | 20.87 | | | | | | | | | | | | | | |
| Sex | 1.51 | .50 | .04 | | | | | | | | | | | | | |
| Age | 39.88 | 11.62 | -.09 | -.02 | | | | | | | | | | | | |
| Ed.years | 1.46 | .50 | .01 | -.03 | -.05 | | | | | | | | | | | |
| GK | 6.95 | 2.43 | -.07 | -.07 | .16** | .02 | | | | | | | | | | |
| Relig. | 2.29 | 2.91 | .19*** | .08 | .13* | -.06 | -.04 | | | | | | | | | |
| Politics | 5.62 | 2.42 | -.15** | .03 | -.15** | .17** | .10 | -.17** | | | | | | | | |
| Agreeable. | 15.58 | 2.61 | -.02 | .33*** | .10* | .04 | .06 | .14** | .18*** | | | | | | | |
| Conscient. | 13.42 | 2.95 | .09 | .00 | .00 | -.01 | .06 | .09 | -.12* | .04 | | | | | | |
| Extrav. | 11.08 | 3.43 | .10* | .03 | .05 | -.03 | .09 | .15** | -.02 | .25** | .10* | | | | | |
| Open. | 14.45 | 2.94 | -.04 | -.08 | -.05 | .08 | .12* | -.07 | .23** | .20** | .07 | .17*** | | | | |
| Neurot. | 11.95 | 3.29 | .10* | .26*** | -.23**' | .15** | -.15** | -.01 | .12* | .08 | -.31*** | -.19*** | -.03 | | | |
| Cl A | 27.59 | 7.00 | .35*** | .01 | -.19** | .05 | -.11* | .03 | -.06 | -.19** | -.20*** | -.29*** | -.03 | .47*** | | |
| Cl B | 34.75 | 7.41 | .28*** | -.05 | -.27** | .08 | .02 | .04 | .04 | -.08 | -.26*** | .15** | .12* | .43*** | .54*** | |
| Cl C | 30.26 | 6.7 | .16** | .08 | -.24** | .11* | -.04 | -.06 | .14** | -.09 | -.23*** | -.39*** | -.07 | .54*** | .73*** | .52*** |

*$p < .05$
**$p < .01$
***$p < 001$.

AdjR$^2$ = .23 with three PDs being significant: Borderline (Beta = .18, t = 2.12, p < .05) Dependent (Beta =.-18, t = 2.54, p < .01) and Schizotypal (Beta = .25, t = 3.84, p < .01).

We repeated the same analyses on the other measure of CT. Table 3 shows the correlational results. Eight of the correlations were significant, showing that higher CT were associated with: Not having a degree, being more religious and politically conservative, being Extraverted and Neurotic, as well as having higher scores on Clusters A, B and C.

We did the same regression: See Table 4, acknowledging the same problems. This indicated that less well educated, religious Conscientious, Extraverts with high scores on Cluster A and B, low on Cluster C were more likely to endorse the CT theories. The regression accounted for almost exactly the same amount of variance as in the other regression.

Again, we did the same hierarchical regression but using all 14 PDs rather than clusters. The result was significant (F(25,332) = 5.42, AdjR$^2$ = .24 with three PDs being significant: Avoidant (Beta = -.28, t = 2.65, p < .01), Passive-Aggressive (Beta = .15, t = 1.93, p < .01) and Schizotypal (Beta = .27, t = 4.19, p < .01).

The two measures of CT (thinking and beliefs) correlate strongly (.58), but our predictor variables do not seem to part in major ways in their effects on predicting CT1 or CT2.

The clinical clusters remain the most important predictors of CT, rendering the other variables insignificant or unimportant. We therefore proceeded to analyse the semantic relationships between the 70 items of the SCATI and the 10 items of CT1 that explicate the conspiracy theories, a total of 80 items. The analysis was made on the general LSA service available at the website lsa.colorado.edu, using the doc-to-doc procedure with 300 factors, similar to the procedure used by Nimon & al. in 2016 [66]. The ensuing output yields a pairwise comparison of overlap in meaning between the 80 items, a symmetric matrix of 3,160 unique pairs that previous studies has shown to be predictive of psychometric characteristics [62,67,68]. This matrix, termed a "semantic similarity matrix" [67] can be viewed as representing the mere overlap in semantic meaning between the items, i.e., purely cognitive features. It does not contain any

Table 4. Regression results for CT2.

|  | B | Std. Error | Std. Beta | t |
|---|---|---|---|---|
| Sex | 1.495 | 2.239 | .036 | .667 |
| Age | -.066 | .093 | -.038 | -.718 |
| School years | -.333 | .242 | -.068 | -1.371 |
| religious | .581 | .370 | .080 | 1.569 |
| Politics | -.367 | .456 | -.044 | -.805 |
| GK | -.167 | .490 | -.017 | -.341 |
| Agreeableness | .195 | .476 | .024 | .410 |
| Conscientiousness | .967 | .381 | .134 | 2.536* |
| Extraversion | .803 | .383 | .133 | 2.097* |
| Openness | -.425 | .382 | -.060 | -1.112 |
| Neuroticism | -.250 | .407 | -.039 | -.615 |
| Cluster A | 1.345 | .240 | .449 | 5.593*** |
| Cluster B | .488 | .208 | .171 | 2.347* |
| Cluster C | -.604 | .277 | -.192 | -2.178* |

$^*p < .05$
$^{**}p < .01$
$^{***}p < 001.$

knowledge about respondent ratings or other features of the scales, but it is structurally comparable to the correlation matrix of the item scores [67].

Our first analysis aimed to check if the semantic information was sufficient to determine the factor structure of the items used in SCATI and CT1. We subjected the semantic similarity matrix to a principal component analysis with oblimin rotation, asking for a total of 15 factors (14 clinical scales plus the CT1). The analysis returned 15 factors with root mean square of the residuals (RMSR) being .03. Uniquely high factor loadings were identifiable for the CT1 items and 13 of the 14 clinical scales, the only scale failing to appear as separate from the other clinical scales was Histrionic. A t-test indicated that the identified unique factors were significantly different from the cross loadings (p < .001). We therefore concluded that the semantic properties of the scales were sufficient to identify the single scales and their relationships with CT1, see Table 5:

As shown in Table 5, the SCATI and the CT1 items stand out as semantically separate in the principal component analysis based on semantics alone. We next went to calculate the average semantic relationships between the clinical scales and our dependent variable, CT1. This is done by averaging the semantically generated cosines in all item-to-item relationships in the matrix. Since each SCATI scales consists of 5 items, their mutual relationships are made of 5*5 = 25 unique item pairs. The relationships between each SCATI scales and the CT1 likewise consist of 5*10 = 50 unique item pairs. The averages of these relationships are calculated and presented in Table 6, along with the empirically observed correlations based on our sampled responses:

Table 6 shows that the semantic characteristics of the SCATI items that are most strongly related to beliefs in conspiracy items also predict the order in which the SCATI scales are correlated with CT1. H9 is supported in that the semantic (i.e., cognitive) properties of the SCATI items are indicative of the strength of their correlations with CT beliefs. The semantic relationships between conspiracy theories and the two SCATI scales schizotypal and paranoia are significantly stronger (p < .05) than those of the other scales.

**Table 5. Semantically generated principal component analysis.** Factor loading for each unique scale ("focus factors") compared to cross-loadings ("orbiting factors") and p-values for the differences.

| Scale | Focus factor loadings | Orbit (cross-loadings) | P-value for the difference |
|---|---|---|---|
| ANTISOCIAL | .25 | .06 | .043 |
| AVOIDANT | .11 | .08 | .696 |
| BORDERLINE | .25 | .07 | .015 |
| DEPENDENT | .22 | .06 | .069 |
| DEPRESSIVE | .13 | .08 | .512 |
| HISTRIONIC | .02 | .09 | .264 |
| NARCISSISTIC | .15 | .08 | .234 |
| OBSESS.-COMPULS. | .26 | .07 | .006 |
| PARANOID | .31 | .05 | .001 |
| PASSIVE-AGGRESSIVE | .23 | .08 | .023 |
| SADISTIC | .31 | .06 | .001 |
| SCHIZOID | .15 | .08 | .360 |
| SCHIZOTYPAL | .30 | .06 | .002 |
| SELF-DEFEATING | .31 | .06 | .001 |
| **CT1** | **.29** | **.03** | **.000** |
| **Full study** | **.22** | **.07** | **.000** |

## Discussion

This study set out to explore how demographics, education, ideology, normal personality traits, and indications of personality disorders are contributing to beliefs in conspiration theories (CTs). Multiple regression found a weak, negative relationship between educational level and CTs, giving modest support to H1. General knowledge (that we used as a proxy for intelligence) was not directly related to CTs and H2 was thereby not supported. Religious beliefs were correlated with CT2 but not CT1, and the effects of religiousness waned in multiple regression, disconfirming H3. Political belongingness did correlate significantly with both measures of CT, but in both cases the effects disappeared in multiple regression, disconfirming H4. The Big Five personality traits were also correlated with CTs but only conscientiousness and extraversion retained significance in the multiple regression models. This finding disconfirmed H5 and H6. The Personality Disorders did show up as main predictors of belief in both CTs. However, we predicted clusters A and B, whereas clusters A and C emerged as significant for both CT1 and CT2. Finally, the latent semantic analysis (LSA) showed a significant relationship between the cognitive properties of the PD scales and belief in CT, in particular for schizotypal disorder. H9 was thereby supported.

The pattern of confirmed vs disconfirmed hypotheses suggests that beliefs in conspiracy theories are related to ideology or general levels of knowledge in terms of superficial correlations, but that these relationships tend to blur when matched against traits measuring personality disorders. These effects should not be taken to indicate that all people who believe in CTs are also showing a personality disorder (PD). Rather, it is likely that personality factors, knowledge levels and ideologically conducive environments contribute to the adoption of CTs, where full-fledged embracement of these may take on the character of personality disorders or be exacerbated by underlying tendences towards such.

This interpretation is supported by the fact that our sample does not on average display elevated scores on the SCATI compared to the clinically established norms. More importantly, not all PDs seem to predispose people to the same degree of CT endorsement. The PDs most strongly related to conspiracy beliefs are the schizotypal and paranoid subtypes. These groups

**Table 6. Semantic relationships (in cosines) between SCATI semantics and CT1, compared with the empirically observed correlations from human responses (bottom line).**

| | ANTI-SOC. | AVOID. | BORD. | DEPEN. | DEPRES. | HISTRI. | NARC. | OBSES. | PARAN. | PASSIV. | SADIST. | SCHIZOID | SCHIZO-TYP. | SELF-DEF. |
|---|---|---|---|---|---|---|---|---|---|---|---|---|---|---|
| AVOIDANT | .45 | | | | | | | | | | | | | |
| BORDERLINE | .47 | .49 | | | | | | | | | | | | |
| DEPENDENT | .45 | .42 | .43 | | | | | | | | | | | |
| DEPRESSIVE | .44 | .50 | .50 | .44 | | | | | | | | | | |
| HISTRIONIC | .40 | .41 | .41 | .40 | .43 | | | | | | | | | |
| NARCISSISTIC | .43 | .47 | .48 | .42 | .50 | .40 | | | | | | | | |
| OBSESSIVE | .52 | .49 | .51 | .46 | .47 | .44 | .46 | | | | | | | | |
| PARANOID | .53 | .53 | .52 | .49 | .51 | .46 | .50 | .56 | | | | | | |
| PASSIVE-AGGR. | .48 | .50 | .50 | .46 | .47 | .42 | .48 | .52 | .58 | | | | | |
| SADISTIC | .37 | .35 | .38 | .33 | .38 | .32 | .36 | .37 | .42 | .37 | | | | |
| SCHIZOID | .29 | .33 | .34 | .30 | .30 | .28 | .32 | .30 | .32 | .29 | .24 | | | |
| SCHIZOTYPAL | .43 | .43 | .41 | .44 | .41 | .38 | .44 | .45 | .51 | .48 | .32 | .30 | | |
| SELFDEFEATING | .38 | .45 | .40 | .41 | .49 | .35 | .47 | .37 | .44 | .46 | .32 | .27 | .39 | |
| **CT1** | *.07* | *.06* | *.05* | *.06* | *.03* | *.04* | *.06* | *.05* | ***.08*** | *.07* | *.06* | *.04* | ***.09*** | *.05* |
| ***Empirical Corr.**** | *.24* | *.07* | *.19* | *.11* | *.14* | *.15* | *.25* | *.20* | ***.29*** | *.22* | *.24* | *.18* | ***.41*** | *.21* |

**r = .70***

*The semantic relationships with clinical scales are correlated .70 (rank-order, p < .01) with the observed correlational equivalents.

could be characterized with affinity to loose associations, cognitive oddities and propensities towards delusion. On the opposite side of the spectrum are the avoidant, depressive and histrionic disorders who might simply be scared of thoughts with conspiracy content and react with different defense mechanism to being exposed to such ideas.

These findings are in line with previous research on the cognitive characteristics of schizophrenic and schizotypal disorders and the diagnostic value of semantic algorithms in detecting such cognitive propensities [19,42,49].

It is thus possible that the manifest agreement on conspiracy theories in groups of people is caused partly by social psychological mechanisms that drive people together in social networks where these ideas are being proposed, and partly by individuals with personality disorders who are willing and capable of propagating such theories with great conviction. Such a propagation of CTs would be a two-thronged approach where vulnerable individuals lend credibility to unlikely ideas that penetrate the capacity for critical thinking in less disturbed but otherwise socially committed network members.

There are three interesting issues that arise from this study both methodological and theoretical. The *first* refers to the generalisability of findings given the measures used. Any researcher is being increasingly faced with an alarming increase in CTs as a range of global crises unfold [24] and may ask the question about the specificity of particular CTs. That is, are the causes, consequences and correlates of beliefs in any/all CTs unique to that CT or generalisable? We are one of very few studies which used more than one, though inevitably strongly positively correlated, measures of beliefs in CTs to check the reliability of our findings. Our findings seem to suggest that there are generalisable aspects to the CTs in the sense that the scales themselves have some internal consistency, they are substantially correlated between themselves, and they seem to be related along similar lines to variables such as ideology, education, and personality traits. Maybe in an even more substantially way, the conspiracy theories all seem related to the same type of cognitive structures as identified through the semantic algorithms. This analysis shows quite clearly the relationship of ideation and cognition between conspiracy theories, paranoia, and schizotypal thinking. This finding relates in an interesting way to previous findings on schizophrenia [19].

The *second* issue concerns the variance explained in the many studies of this kind [4]. Many researchers have noted that although they assessed a wide variety of individual difference measures (biography, demography, ideology, personality) they could not account for very much of the variance (often less than 10%) which was often inflated by method invariance. The question asked was what other individual difference factors, not assessed, may account for these beliefs? In this study we measured a number of individual variables which accounted for around 20% of the variance in both analyses. We believe that what is unique about this paper is the number of predictor variables that we used at the same time to explore correlates of beliefs in CTs.

The *third* issue, is how people integrate specific CTs into their general CT world view such that when asking them about a number of them, as done in this study, there is high alpha indicating internal reliability. In this study there was a difference between the two alphas: .60 vs .91. The relatively low alpha of the first Conspiracy thinking measure may have resulted from some relatively "extreme" ideas being tested such as alien interventions. Interestingly Walter and Drochon [2] found only one factor when they factor analysed their data set but did note that their results left room for at least two complementary approaches namely determinants of, and consequences of, *generic* conspiracy thinking vs *specific* individual conspiracy beliefs. It is possible that they are some theories such as the "ancient alien astronaut theory" which only very few are able to endorse and integrate into their conspiracist world view [69].

## Limitations and suggestions for further research

Most of the psychological research in this area has confirmed the hypotheses about what sort of people are more likely to hold conspiracist beliefs. Further they offer plausible explanations about why they do so. Yet we know little about the process. For instance, how do people hear about CTs: do they seek out media outlets that endorse these theories such as very specific television channels and website? Do they join, or shun, groups of believers? How do they answer their critics, and how much have they had to do so? Which are their most central CT beliefs, and how well integrated are these with their socio-political world view?

It would also be interesting to note how CT beliefs are related to other superstitious, magical or paranormal beliefs as well as attitudes to science and rationality. For instance, are those interested in, and believers concerning astrology, more likely to accept CT? Are people more skeptical about conventional medicine and attracted to alternative medicine more interested in CT?

Like all others this study had limitations. The population was skewed towards better educated middle class people from one country. They may be less inclined to accept certain CTs, but possibly more receptive to others. It is clearly always best to try to have a large, diverse and representative population in such studies. Our test of General Knowledge (GK) was very short. Though we believe it to be both reliable and valid, we see that a more traditional and encompassing IQ test measuring g (general intelligence) could yield better and more credible information about the effect of cognitive ability on CTs. Further, we had very brief measures of religious and political beliefs which were related to CTs and it would be interesting to explore these associations with more comprehensive measures of those beliefs, as well as knowledge about the respondents' level of commitment to, and involvement in their ideological communities. For instance membership of, and interaction with, particular religious and political groups, may say a lot about beliefs in CTs over time.

The internal reliability of one CT measure was lower than the usual acceptable cut-off of .7. Next, as always many of our correlations, even though significant were low. Those shown in Tables 1 and 3 are not Bonferroni corrected which means many are not significant. This shows yet again the search for individual difference variables which explain belief in CT remain unknown, and which may throw light on the psychological function of holding CTs.

## Supporting information

**S1 File.**
(SAV)

## Author Contributions

**Conceptualization:** Adrian Furnham.

**Data curation:** Charlotte Robinson.

**Formal analysis:** Jan Ketil Arnulf, Charlotte Robinson, Adrian Furnham.

**Methodology:** Adrian Furnham.

**Resources:** Adrian Furnham.

**Validation:** Jan Ketil Arnulf.

**Writing – original draft:** Jan Ketil Arnulf, Adrian Furnham.

**Writing – review & editing:** Jan Ketil Arnulf.

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
