## [Decision Letter · Decision Letter 0]

17 Feb 2022

PONE-D-21-40074Bright and Dark-Side Personality and Conspiracy ThinkingPLOS ONE

Dear Dr. Arnulf,

Thank you for submitting your manuscript to PLOS ONE. After careful consideration, we feel that it has merit but does not fully meet PLOS ONE’s publication criteria as it currently stands. Therefore, we invite you to submit a revised version of the manuscript that addresses the points raised during the review process.

I have obtained reviews from two competent reviewers. Please carefully read their comments.

I will start with my own.

The hierarchical regression analysis of which results are presented in Table 9 is highly problematic. First of all, the way you had constructed dependent variable (factor with loadings on conspiracy variables, based on the factor analysis of the overall set of variables) was not appropriate. The reason is that it contains - to an extent – the variance of the variables that you were using as predictors later in this regression analysis (this is the reason why your multiple correlation is so inflated in this analysis). Moreover, the rationale for this analysis is totally unclear: you showed the results with conspiratorial thinking and conspiratorial beliefs separately (I see the logic in this – I guess to further check similarity of these two by projecting them in the space defined by the same predictor set), and then, you are showing practically the same thing with a variable that is some unfortunate composite of these two (and the rest of the variables included in this FA). I would highly recommend to remove this analysis, the text attached to it, as well as Figure 1.

A little can be achieved by stretching the notion of dark-of side personality on cluster A. Cluster A is implicated in conspiracy beliefs because it measures psychotic-like phenomena, not because of its darkness (if dark side of personality has anything in common with what is measured by dark triad or dark tetrad scales, i.e. malevolent personality). In other words, If you partial out schizotypy from Cluster A it would cease to predict anything conspiratorial (beliefs, thinking, mind-set). Cluster B (more related to the notion of dark personality) is obviously of secondary importance, even part of its relatedness to conspiracy is due to the correlation with Cluster A. Therefore, my recommendation would be to take into consideration the first suggestion of r#2 very seriously. I would recommend to rely on the usual use of the notion of darkness in this field. Your own results revealed much more peculiarity in conspiracy than darkness.

You did not find support neither for H2 nor H5. Please, be more careful with the interpretation of your own results.

Your explanation of the mediator role of political ideology between conscientiousness and conspiracy does not seem to be convincing: openness and agreeableness are much more implicated in political ideology than conscientiousness.

Variable "degree" is not explained (Years of schooling? Educational levels (if levels, how many)?). In regressions with CT1 and CT2 as Dvs, it looks like higher level of education predicts conspiracy, not the other way around. It is strange that your educational levels start with higher levels and end with lower levels (years) of education. Please, be careful with this - I suggest higher levels of this variable to reflect higher education level.

Minor – rename Table 1 as Table 2.

Please, avoid using term interaction in describing your H9, you did not test the interaction between personality and ideology.

Please, use the same labels throughout the article (e.g. BCTI but not CT2, or CT2 not BCTI, or use both).

We look forward to receiving your revised manuscript.

Kind regards,

Goran Knezevic

Academic Editor

PLOS ONE

Journal Requirements:

a) Did participants provide their written or verbal informed consent to participate in this study?

4. Please include a copy of Table 5 which you refer to in your text on page 14.

Reviewers' comments:

Reviewer's Responses to Questions

**Comments to the Author**

1. Is the manuscript technically sound, and do the data support the conclusions?

Reviewer #1: No

Reviewer #2: Yes

2. Has the statistical analysis been performed appropriately and rigorously? 

Reviewer #1: I Don't Know

Reviewer #2: Yes

3. Have the authors made all data underlying the findings in their manuscript fully available?

Reviewer #1: Yes

Reviewer #2: No

4. Is the manuscript presented in an intelligible fashion and written in standard English?

Reviewer #1: Yes

Reviewer #2: Yes

5. Review Comments to the Author

Reviewer #1: This paper presents findings from an online study of 397 community residents ages 19-71 most of whom were British in nationality. They completed an extensive battery of personality, political, and other demographic differences along with two measures of acceptance of conspiracies, a more general measure and one assessing belief in specific conspiracies. The two measures of conspiratorial thinking were moderately correlated (r=.58), which is about what one would expect given the respective levels of reliability (.60 and .91). Regression analyses using the various correlates produced similar profiles of associations, and a factor model placed the two measures in roughly the same location in a two-dimensional space, defined by political ideology versus measures of personality disorders.

The findings show that both types of measures of conspiratorial thinking are largely related in the same ways to various characteristics. There are more of those for the specific conspiracies, but that is not explained, and so I’m not sure there is much new here other than that. The authors suggest that they are testing the hypothesis that “personality and ideological beliefs interact to explain a common pathway toward CT.” (page 6). But I see no effort to test that hypothesis as all of the analyses involve tests of single correlates. There is evidence that after controlling for the personality disorders, the relation with political ideology disappears. But I don’t see how that tests the hypothesis as it is stated. Apparently, persons with those disorders in this sample are more likely to be on the conservative end of the political spectrum. The authors claim that their findings support the prediction, but I don’t how they can say that other than that is what they are reading into the pattern of correlates.

The paper would benefit a lot from a more discerning discussion of what the findings say and what questions they answer.

There is some confusion at the bottom of page 13, when the authors say that various characteristics were related to endorsement of CT theories: Extraverts with high scores on Cluster A and B, low on Cluster C were more likely to endorse CT theories.” But that is not what the regressions tell us. Those are independent predictors. And does CT include theories?

Reviewer #2: The paper is very well written, concise, and clear. The research problem is very relevant and actual and the strength of this study is in the involvement of different variables in the prediction of CT. However, there are some concerns and suggestions for improvement of the manuscript:

- The title is not appropriate since the authors explored basic and psychopathological traits and not traits that are referred to the bright and dark side of personality in common sense (Dark Triad, Light Triad). Maybe title such is: Dispositional and ideological factors of conspiracy thinking and beliefs would be more appropriate.

- I am not sure what insight we get with separate analyses on CT1 and CT2. The final analysis presented in Table 9 seems that subsumed previous results and it would be economic to keep only these results since two conspiracy measures are already highly correlated. In the introduction, the authors did not explain differences between CT1 and CT2, nor their different correlates, thus separate analysis is not argumented. If they want to keep it (with more elaboration and arguments for that in Introduction), then in the 1st step of prediction of CT1, CT2 should be entered, and vice versa.

The other concern in this part is regarding stepwise regression analysis which is very criticized and should be avoided. Instead, please do a regular, enter regression. If there is a problem with multicollinearity you could try to merge some variables in composite scores if that is justified.

- The other main concerns in regarding General Knowledge Test. It is not clear whether the authors shortened the test for this study or short version of the test is already validated elsewhere? Please report alpha for the used tests in the Method section and not in the Discussion. Furthermore, authors identify GKT with crystallized int. but the correlations between GKT with Openness are the same as with fluency test, while fluency test does not correlate sig. with Openness. I think that authors will need to be more careful regarding the conclusion about relations between crystall. int. and conspiracy theories since the used measure (GKT) and general knowledge as a construct per se is a construct between personality and intelligence. The conclusion should be more in line with what GKT really measures. Also, IQ in all tables and results should be replaced with GKT.

- The joint factor analysis (p. 14) is not argumented in Introduction and it is not in line with aims. Please delete this part.

- In discussion, there is not much explanation of the results, for example, why low N and high C are associated with CT and not A and O as expected?

Minor points:

- p.3 „Indeed, there are good theoretical reasons to believe that the other disorders (i.e. schizotypy) are more related to CTs than the two assessed in the dark triad [18].“ – please correct this because Dark Triad is subclinical and not clinical construct. The traits in DT are not considered disorders.

- there is a lack of reference in some cases, e.g., in a statement such as „based on the previous literature“

- title in Table 1 is missing and please add a note in all tables, what is CT1 and CT2

- I am not sure what variable type is a degree (binary or ordinal), please specify in the sample description. If it is binary, add in a note how it is coded. It is somewhat odd to state „not having a degree“ at p.13.

- p.14 please check this: Passive-Aggressive (Beta=..15,

- delete Fig. 1, everything is already stated in Table 9.

- p.19 authors stated „two studies“ but this is one study, corr. and reg. analysis belong to the one study

- p.17 „allegiance to an ingroup (through conscientiousness)“, allegiance to an ingroup is more Agreeableness and not Conscientiousness

- p.17 „Thus, for instance, less intelligent and well-educated extraverts and neurotics may be less or more prone to holding theories than stable introverts“ I am not sure how much is this realistic since education and int. are correlated.

- p.18 the suggestions for future studies are something that is already very well documented, e.g. „It would also be interesting to note how CT beliefs are related to other superstitious, magical or paranormal beliefs as well as attitudes to science and rationality.“

- since the authors stated „Those shown in Tables 1 and 3 are not Bonferroni corrected which means many are not significant“ the question is why they did not use p-correction already? In any case, it is recommended to use p-correction, thus authors should use it here.

6. PLOS authors have the option to publish the peer review history of their article (what does this mean?). If published, this will include your full peer review and any attached files.

Reviewer #1: No

Reviewer #2: **Yes: **Bojana M. Dinić

---

## [Author Response · Author response to Decision Letter 0]

30 Mar 2022

Subject: PLOS ONE Decision: Revision required [PONE-D-21-40074] - [EMID:2f14363b9625c539]

PONE-D-21-40074

Bright and Dark-Side Personality and Conspiracy Thinking

PLOS ONE

Dear Dr. Knezevic

RESPONSE: Thank you for email concerning the above paper and attaching reviewers comments. I note below how we have responded to each.

Your Comments

The hierarchical regression analysis of which results are presented in Table 9 is highly problematic. First of all, the way you had constructed dependent variable (factor with loadings on conspiracy variables, based on the factor analysis of the overall set of variables) was not appropriate. The reason is that it contains - to an extent – the variance of the variables that you were using as predictors later in this regression analysis (this is the reason why your multiple correlation is so inflated in this analysis). Moreover, the rationale for this analysis is totally unclear: you showed the results with conspiratorial thinking and conspiratorial beliefs separately (I see the logic in this – I guess to further check similarity of these two by projecting them in the space defined by the same predictor set), and then, you are showing practically the same thing with a variable that is some unfortunate composite of these two (and the rest of the variables included in this FA). I would highly recommend to remove this analysis, the text attached to it, as well as Figure 1. 

RESPONSE: THIS PART IS NOW DELETED ENTIRELY

A little can be achieved by stretching the notion of dark-of side personality on cluster A. Cluster A is implicated in conspiracy beliefs because it measures psychotic-like phenomena, not because of its darkness (if dark side of personality has anything in common with what is measured by dark triad or dark tetrad scales, i.e. malevolent personality). In other words, If you partial out schizotypy from Cluster A it would cease to predict anything conspiratorial (beliefs, thinking, mind-set). Cluster B (more related to the notion of dark personality) is obviously of secondary importance, even part of its relatedness to conspiracy is due to the correlation with Cluster A. Therefore, my recommendation would be to take into consideration the first suggestion of r#2 very seriously. I would recommend to rely on the usual use of the notion of darkness in this field. Your own results revealed much more peculiarity in conspiracy than darkness. 

RESPONSE: THE DARKNESS PART IS NOW LARGELY DELETED FROM THE DISCUSSION AND THE FOCUS IS ON THE CLINICAL SCALES AND CLUSTERS.

You did not find support neither for H2 nor H5. Please, be more careful with the interpretation of your own results. 

RESPONSE: ALL HYPOTHESES ARE NOW RESTATED AND THEIR EMPIRICAL STATUS COMMENTED IN A WRAP-UP AT THE BEGINNING OF THE DISCUSSION.

Your explanation of the mediator role of political ideology between conscientiousness and conspiracy does not seem to be convincing: openness and agreeableness are much more implicated in political ideology than conscientiousness.

RESPONSE: WE HAVE DELETED THIS PART OF THE ANALYSIS AND INSTEAD FOCUSED ON THE CONTRIBUTIONS OF THE PERSONALITY DISORDER CATEGORIES IN MULTIPLE REGRESSION.

Variable "degree" is not explained (Years of schooling? Educational levels (if levels, how many)?). In regressions with CT1 and CT2 as Dvs, it looks like higher level of education predicts conspiracy, not the other way around. It is strange that your educational levels start with higher levels and end with lower levels (years) of education. Please, be careful with this - I suggest higher levels of this variable to reflect higher education level. 

RESPONSE: WE PREVIOUSLY USED A REVERSE CODED, DICHOTOMOUS VARIABLE. NOW USING THE NUMBERS OF YEARS OF EDUCATION IN THE EXPECTED DIRECTION (HIGHER NUMBERS = MORE YEARS OF EDUCATION).

Minor – rename Table 1 as Table 2. 

RESPONSE: DONE AS REQUESTED

Please, avoid using term interaction in describing your H9, you did not test the interaction between personality and ideology. 

RESPONSE: NOW DELETED AS SUGGESTED

Please, use the same labels throughout the article (e.g. BCTI but not CT2, or CT2 not BCTI, or use both). 

RESPONSE: DONE AS REQUESTED

RESPONSE: HOPEFULLY THIS HAS BEEN DONE

2. Please amend your current ethics statement to address the following concerns: a) Did participants provide their written or verbal informed consent to participate in this study? b) If consent was verbal, please explain i) why written consent was not obtained, ii) how you documented participant consent, and iii) whether the ethics committees/IRB approved this consent procedure. 

RESPONSE: WE USED THE STANDARD PROCEDURE FOR PROLIFIC. THE ETHICS COMMITTEE CONSENTED TO PARTICIPANTS AGREEING TO HAVE THEIR ANONYMISED RESPONSES AGGREGATED, ANALYSED AND RESPONSE: SUBSEQUENTLY PUBLISHED. WE HAVE DONE THIS WITH NUMEROUS STUDIES.

RESPONSE: WE CONSISTENTLY NOTE THAT IF ANYONE WISHES TO SEE THE DATA WE WILL SEND THEM AN SPSS FILE WITH THE VARIABLES EXPLAINED. THE FILE IS COMPLICATED AND HAS NUMBER OF VARIABLES THAT WE DID NOT ANALYSE OR REPORT FOR THIS STUDY

3. Please include a copy of Table 5 which you refer to in your text on page 14.

RESPONSE: TABLES REVISED AND EDITED.

Reviewer #1: 

This paper presents findings from an online study of 397 community residents ages 19-71 most of whom were British in nationality. They completed an extensive battery of personality, political, and other demographic differences along with two measures of acceptance of conspiracies, a more general measure and one assessing belief in specific conspiracies. The two measures of conspiratorial thinking were moderately correlated (r=.58), which is about what one would expect given the respective levels of reliability (.60 and .91). Regression analyses using the various correlates produced similar profiles of associations, and a factor model placed the two measures in roughly the same location in a two-dimensional space, defined by political ideology versus measures of personality disorders.

RESPONSE: A FAIR SUMMARY

The findings show that both types of measures of conspiratorial thinking are largely related in the same ways to various characteristics. There are more of those for the specific conspiracies, but that is not explained, and so I’m not sure there is much new here other than that. The authors suggest that they are testing the hypothesis that “personality and ideological beliefs interact to explain a common pathway toward CT.” (page 6). But I see no effort to test that hypothesis as all of the analyses involve tests of single correlates. There is evidence that after controlling for the personality disorders, the relation with political ideology disappears. But I don’t see how that tests the hypothesis as it is stated. 

RESPONSE: WE BELIEVE THAT AS WE HAVE RESTATED THE HYPOTHESES AND REWRITTEN MUCH OF THE DISCUSSION WE HAVE NEW AND INTERESTING FINDINGS

Apparently, persons with those disorders in this sample are more likely to be on the conservative end of the political spectrum. The authors claim that their findings support the prediction, but I don’t how they can say that other than that is what they are reading into the pattern of correlates.

RESPONSE: WE HAVE ATTEMPTED AN EXPLANATION OF THIS POINT

The paper would benefit a lot from a more discerning discussion of what the findings say and what questions they answer.

RESPONSE: WE BELIEVE THAT WE HAVE NOW DONE THIS

There is some confusion at the bottom of page 13, when the authors say that various characteristics were related to endorsement of CT theories: Extraverts with high scores on Cluster A and B, low on Cluster C were more likely to endorse CT theories.” But that is not what the regressions tell us. Those are independent predictors. And does CT include theories?

RESPONSE: THANK YOU. WE HAVE RE-VISTED AND CORRECTED THIS POINT

Reviewer #2:

 The paper is very well written, concise, and clear. The research problem is very relevant and actual and the strength of this study is in the involvement of different variables in the prediction of CT.

RESPONSE: EXCELLENT NEWS

 However, there are some concerns and suggestions for improvement of the manuscript:

- The title is not appropriate since the authors explored basic and psychopathological traits and not traits that are referred to the bright and dark side of personality in common sense (Dark Triad, Light Triad). Maybe title such is: Dispositional and ideological factors of conspiracy thinking and beliefs would be more appropriate. 

 RESPONSE: EXCELLENT SUGGESTION THANK YOU, NOW CHANGED

- I am not sure what insight we get with separate analyses on CT1 and CT2. The final analysis presented in Table 9 seems that subsumed previous results and it would be economic to keep only these results since two conspiracy measures are already highly correlated. In the introduction, the authors did not explain differences between CT1 and CT2, nor their different correlates, thus separate analysis is not argumented. If they want to keep it (with more elaboration and arguments for that in Introduction), then in the 1st step of prediction of CT1, CT2 should be entered, and vice versa. 

RESPONSE: WE BELIEVE THAT OPERATING WITH TWO MEASURES OF CT ADDS STRENGTH TO OUR CLAIM THAT THERE EXISTS A GENERLISABLE TENDENCY TO ADOPT CTS IN MANY FORMS, AND THAT THIS GENERALISABLE TENDENCY IS LINKED TO THE PSYCHOLOGICAL MECHANISMS THAT WE OUTLINE.

 THESE IDEAS ARE HOPFULLY NOW BETTER EXPLAINED IN THE TEXT.

The other concern in this part is regarding stepwise regression analysis which is very criticized and should be avoided. Instead, please do a regular, enter regression. If there is a problem with multicollinearity you could try to merge some variables in composite scores if that is justified. 

RESPONSE: THE STEPWISE REGRESSION IS NOW DELETED.

- The other main concerns in regarding General Knowledge Test. It is not clear whether the authors shortened the test for this study or short version of the test is already validated elsewhere? Please report alpha for the used tests in the Method section and not in the Discussion. Furthermore, authors identify GKT with crystallized int. but the correlations between GKT with Openness are the same as with fluency test, while fluency test does not correlate sig. with Openness. I think that authors will need to be more careful regarding the conclusion about relations between crystall. int. and conspiracy theories since the used measure (GKT) and general knowledge as a construct per se is a construct between personality and intelligence. The conclusion should be more in line with what GKT really measures. Also, IQ in all tables and results should be replaced with GKT. 

RESPONSE: WE HAVE DELETED REFEFENCES TO IQ AND REPLACED THEM WITH THE ABBREVIATION GK INSTEAD.

- The joint factor analysis (p. 14) is not argumented in Introduction and it is not in line with aims. Please delete this part. 

RESPONSE: DELETED AS SUGGESTED

- In discussion, there is not much explanation of the results, for example, why low N and high C are associated with CT and not A and O as expected? 

RESPONSE: THE DISCUSSION IS NOW RE-DESIGNED TO EXAMINE THE REGRESSION RESULTS THAT FOCUS ON THE PERSONALITY DISORDERS.

Minor points:

- p.3 „Indeed, there are good theoretical reasons to believe that the other disorders (i.e. schizotypy) are more related to CTs than the two assessed in the dark triad [18].“ – please correct this because Dark Triad is subclinical and not clinical construct. The traits in DT are not considered disorders. 

RESPONSE: REFERENCES TO THE DARK TRIAD ARE NOW TAKEN OUT OF THE CORE PART OF THE STUDY AND ONLY SPORADICALLY REFERRED TO AS PART OF THE PREVIOUS RESEARCH IN THE FIELD.

RESPONSE: YOU AND THE REVIEWERS HAVE GIVEN US THE OPPORTUNITY TO SIGNIFICANTLY IMPROVE THE CLARITY, FOCUS AND ANALYSIS OF THE PAPER FOR WHICH WE ARE MOST GRATEFUL.

---

## [Decision Letter · Decision Letter 1]

23 May 2022

PONE-D-21-40074R1Dispositional and ideological factor correlate of conspiracy thinking and beliefsPLOS ONE

Dear Dr. Arnulf,

Thank you for submitting your manuscript to PLOS ONE. After careful consideration, we feel that it has merit but does not fully meet PLOS ONE’s publication criteria as it currently stands. Therefore, we invite you to submit a revised version of the manuscript that addresses the points raised during the review process.

The article is now considerably improved. However, both reviewers had further recommendation that I would like you to address properly. While r#2 has more technical suggestions, the objections of r#1 are more general and substantive. The issue of importance and novelty of the findings in the context of existing knowledge should be more elaborated. As emphasized by both reviewers, the semantic analysis requires further explanation and elaborations. As none of us is familiar with the analysis (and probably most of our audience) the method and its rationale should be more elaborated. I am curious how you would respond to the following statement: to the extent that the relationship between any two constructs is based on the similarity of the words used to describe them, the more psychologically trivial is the relationship (i.e., constructs are too close to each other, predictor-criterion overlap is substantial). Apart from reviewers requested, I would suggest to have the values in the Table 6 better explained (Are correlations between PDS and CTs+GK presented in the last three columns?; How exactly sum factor scores from LSA are calculated and why this information is important beyond the three factors loadings?; Isn’t it the case that the correlations between the loadings on the three factors and correlations of PDs and CTs are of importance, not the correlations between the factor loadings and sum of loadings across the factors or their ranks).

We look forward to receiving your revised manuscript.

Kind regards,

Goran Knežević

Academic Editor

PLOS ONE

Reviewers' comments:

Reviewer's Responses to Questions

**Comments to the Author**

1. If the authors have adequately addressed your comments raised in a previous round of review and you feel that this manuscript is now acceptable for publication, you may indicate that here to bypass the “Comments to the Author” section, enter your conflict of interest statement in the “Confidential to Editor” section, and submit your "Accept" recommendation.

Reviewer #1: (No Response)

Reviewer #2: (No Response)

2. Is the manuscript technically sound, and do the data support the conclusions?

Reviewer #1: No

Reviewer #2: Yes

3. Has the statistical analysis been performed appropriately and rigorously? 

Reviewer #1: I Don't Know

Reviewer #2: Yes

4. Have the authors made all data underlying the findings in their manuscript fully available?

Reviewer #1: No

Reviewer #2: No

5. Is the manuscript presented in an intelligible fashion and written in standard English?

Reviewer #1: Yes

Reviewer #2: Yes

6. Review Comments to the Author

Reviewer #1: This revised paper focuses on a semantic analysis of various personality measures and tendencies toward conspiratorial thinking in a convenience sample of mostly British adults. The authors find that various clinical forms of thinking such as schizotypal tendencies are strong correlates of conspiracy thinking. This has been found before and so it seems that what is novel in this paper is the use of semantic analysis to identify why people who endorse those clinical symptoms are more likely to also believe in conspiracies. Publication in Plos One does not require novel findings, but if replication is the goal, then this is a poor sample for that purpose. It is limited to people who are employed, with no obvious justification for that restriction. And it is unlikely to be representative of any population since it is a convenience sample with no evidence of generalizability. The authors argue in the future directions section that the results raise interesting questions, but the ones suggested seem to have been studied. There is evidence that conspiracy believers use non-mainstream media and that they are attracted to alternative medical treatments. All of this prior work should be cited and if the present results add something of value to those findings, then it might be worth saying what that is.

With these concerns aside, there are questions about how the semantic analysis was conducted. How were the inter-relations between the semantic scores subjected to a principal component analysis? I can imagine how that was done, but providing the details would help.

The more important analysis of the relation between the semantic scores and conspiracy beliefs is totally opaque. It is not at all clear what is meant by the sum of factor loadings in Table 6 or how those were related to ratings of conspiracy beliefs. Nor is there any clear interpretation of the second factor that is claimed to correlate with conspiracy thinking.

The authors seem to gloss over the finding that conscientiousness is related to conspiracy thinking. Why is that? Given the unequal reliabilities of the various measures used to assess correlates of conspiracy thinking, regression analysis is a poor guide to detecting true relations. I would go with the simple correlations more so than the regressions.

The authors claim that the two measures of conspiracy mentality are only modestly correlated (.58) but given the reliabilities of the measures of .60 and .91 (as reported in the Discussion when it should be in the Methods), one would only expect a correlation of about .54. So, the correlation that was observed suggests that they two scales are virtually identical.

Reviewer #2: Dear authors,

Thank you for your answers and comments. The manuscript is improved, but there are still some minor issues that should be addressed before the acceptance for publication:

Introduction

- Please add a short explanation of similarities and differences between 2 measures of CTs since you built your case based on these 2 measures.

- Seems like the whole introduction is based on belief in CTs, and not conspiracy thinking. In the description of previous research please specify for each study which CTs measure is used

(specify the name of the measure, i.e., in Walter and Drochon study, etc.). This would contribute to a better understanding of the introduction of 2 measures of CT.

- Although the authors stated that they delete the term cryst. int., it is still present in the hypothesis:

"We also used a short measure of crystalised intelligence to test the hypothesis that higher IQ test scores

would be negatively correlated with belief in CTs (H2)." as well as in the whole text (e.g., in the discussion they used the term intelligence and not cryst. int.). Please specify that you expect negative relations between

the general knowledge as the indicator of cryst. int. and CTs, or just - negative relations between the general knowledge and CTs.

In addition, in this hypothesis, only beliefs in CTs are mentioned and not conspiracy thinking or CTs in general, as in other hypotheses.

Results

- There is no M and SD for CT2 in Table 1, please merge Tables 1 and 3. Please add alphas for all measures in Table 1 and report either sum or mean scores for used instruments

(mean scores are better). Please delete descriptives for sex, it is odd. Add a note with variable explanation in all tables.

- I am not sure why in LSA only CT1 was entered, and not CT2. Please explain it at the beginning of the analysis. Are there some technical issues? I am not familiar with LSA and I could assume that readers would not be also familiar, thus the comment "(CT2 was not part of the analysis at all and the number is

statistically significant even if N (the number of scales) is only 14)" should be explained more clearly.

- Seems like the rotation was not used in PCA, was it on purpose? Also, the statement "Here, the CT1 items have the strongest loadings,

followed by the items from schizotypal scale" is not true, since the Depressive scale had loading -.216.

- The following analysis presented in Table 6 is not clear to me and it is not clear why GK is in the table. The answer on your H9 could be get based on the pattern structure of the factors in Table 5 (with the use of rotation).

If you have other rationale, please explain. Also sentence "If the CT1 factor loadings are removed from the table..." should be changed to "removed from the analysis..."

Discussion

- The discussion lack similarities and differences between CT1 and CT2. In addition, the statement "In this study we measured a number of individual variables which accounted for

22% of the variance in both analyses" should be replaced with - around 20%, since CT1 was explained by 20% and CT2 by 21%.

Other

- Check APA for use of capital letters (it should be personality disorder and conspiracy theories, but Dark Triad and not dark-triad or dark triad)

- Delete (Crystallised IQ) from the explanation of the General Knowledge Test.

- Delete M and SD from the description of the measures since they are already presented in Table 1.

- Results for model testing remove from Table 2 (Model Adj. R2 = .20. F = 6.868, p < .001) and Table 4 (Model Adj. R2 = .21. F = 7.250, p < .001) and report in text as you did for next analysis.

- Please specify on p.10 that you use hierarchical regression analysis.

7. PLOS authors have the option to publish the peer review history of their article (what does this mean?). If published, this will include your full peer review and any attached files.

Reviewer #1: No

Reviewer #2: **Yes: **Bojana M. Dinić

---

## [Author Response · Author response to Decision Letter 1]

7 Jul 2022

Comments to reviewers - PONE-D-21-40074R1

Dispositional and ideological factor correlate of conspiracy thinking and beliefs

PLOS ONE

Dear prof. Knežević!

Please find below our comments and answers to the questions and concerns addressed by yourself and the reviewers: 

The article is now considerably improved. 

Thank you!

However, both reviewers had further recommendation that I would like you to address properly. While r#2 has more technical suggestions, the objections of r#1 are more general and substantive. The issue of importance and novelty of the findings in the context of existing knowledge should be more elaborated. As emphasized by both reviewers, the semantic analysis requires further explanation and elaborations. As none of us is familiar with the analysis (and probably most of our audience) the method and its rationale should be more elaborated. 

The technique is now explicated in further detail at several points in the manuscript. The methodological approach has also been more aligned with traditional statistics tables. We have also added more references to the growing body of research using these methods. Hopefully these passages are more intuitively understandable now.

I am curious how you would respond to the following statement: to the extent that the relationship between any two constructs is based on the similarity of the words used to describe them, the more psychologically trivial is the relationship (i.e., constructs are too close to each other, predictor-criterion overlap is substantial). 

This question is pertinent and interesting. However, it has been thoroughly dealt with in a number of the references in the text, these will be listed again here. The core of the problem is how almost all survey-based correlation matrices seem heavily influenced by semantic relationships. These semantic relationships seem to exist below the usual conventions for scale construction in psychometrics, such as various types of factor analysis, rotation of scales and thresholds for cross-loadings. Thus, several publications using semantic algorithms have shown that the predictor-criterion overlap problem even where the strictest psychometric techniques have been used. There is a growing need to determine the relationship between semantic properties of scales and the influence of attitude strength. In fact, the present manuscript offers an interesting window to the intricate relationship between attitudinal and semantic/cognitive mechanisms involved in survey responses, see table 6 which contains a direct comparison of semantic predicted with empirically observed relationships. The following publications address these questions directly:

The relationship between semantic patterns and attitude strength: Arnulf, J. K., Larsen, K. R., Martinsen, O. L., & Egeland, T. (2018). The failing measurement of attitudes: How semantic determinants of individual survey responses come to replace measures of attitude strength. Behavior Research and Methods, 50(6), 2345-2365. 

The use of text algorithms to replace/complement Likert scales and other types of psychological measures: Kjell, O. N. E., Kjell, K., Garcia, D., & Sikstrom, S. (2019). Semantic Measures: Using Natural Language Processing to Measure, Differentiate, and Describe Psychological Constructs [Article]. Psychological Methods, 24(1), 92-115. https://doi.org/10.1037/met0000191

The pervasiveness of semantically produced relationships in established measurement scales: Arnulf, J. K., Larsen, K. R., Martinsen, O. L., & Bong, C. H. (2014). Predicting survey responses: how and why semantics shape survey statistics on organizational behaviour. PLoS ONE, 9(9), e106361. https://doi.org/10.1371/journal.pone.0106361

How semantic properties of scales may help us model scales and their relationships a priori: Rosenbusch, H., Wanders, F., & Pit, I. L. (2020). The Semantic Scale Network: An online tool to detect semantic overlap of psychological scales and prevent scale redundancies. Psychological Methods, 25, 380–392. https://doi.org/10.1037/met0000244

The philosophical problems of semantically determined scales: Arnulf, J. K., & Larsen, K. R. (2021). Semantic and ontological structures of psychological attributes. In D. Wood, S. J. Read, P. D. Harms, & A. Slaughter (Eds.), Measuring and modeling persons and situations (pp. 69-102). Academic Press. https://doi.org/10.1016/B978-0-12-819200-9.00013-2

Apart from reviewers requested, I would suggest to have the values in the Table 6 better explained (Are correlations between PDS and CTs+GK presented in the last three columns?; How exactly sum factor scores from LSA are calculated and why this information is important beyond the three factors loadings?; Isn’t it the case that the correlations between the loadings on the three factors and correlations of PDs and CTs are of importance, not the correlations between the factor loadings and sum of loadings across the factors or their ranks).

Tables 5 and 6 are now completely changed in ways that comply more with traditional ways of representing data, and with more extensive explanations that we hope will make everything more transparent.

See below

See attached

See attached

Reviewers' comments:

Reviewer's Responses to Questions

Comments to the Author

4. Have the authors made all data underlying the findings in their manuscript fully available?

requires authors to make all data underlying the findings described in their manuscript fully available without restriction, with rare exception (please refer to the Data Availability Statement in the manuscript PDF file). The data should be provided as part of the manuscript or its supporting information, or deposited to a public repository. For example, in addition to summary statistics, the data points behind means, medians and variance measures should be available. If there are restrictions on publicly sharing data—e.g. participant privacy or use of data from a third party—those must be specified. 

All the necessary data will be uploaded with the manuscript

Reviewer #1: No

Reviewer #2: No

5. Is the manuscript presented in an intelligible fashion and written in standard English?

Reviewer #1: Yes

Reviewer #2: Yes

6. Review Comments to the Author

Reviewer #1: This revised paper focuses on a semantic analysis of various personality measures and tendencies toward conspiratorial thinking in a convenience sample of mostly British adults. The authors find that various clinical forms of thinking such as schizotypal tendencies are strong correlates of conspiracy thinking. This has been found before and so it seems that what is novel in this paper is the use of semantic analysis to identify why people who endorse those clinical symptoms are more likely to also believe in conspiracies. Publication in Plos One does not require novel findings, but if replication is the goal, then this is a poor sample for that purpose. It is limited to people who are employed, with no obvious justification for that restriction. And it is unlikely to be representative of any population since it is a convenience sample with no evidence of generalizability. The authors argue in the future directions section that the results raise interesting questions, but the ones suggested seem to have been studied. There is evidence that conspiracy believers use non-mainstream media and that they are attracted to alternative medical treatments. All of this prior work should be cited and if the present results add something of value to those findings, then it might be worth saying what that is.

We believe that our sample of nearly 400 adults drawn from the Prolific platform is a good sample from which one could easily generalize. Indeed, we have published papers in PLOS ONE (2021) using a similar sample from the same provider. We specify employed people to reduce the number of students.

As to the latter point, we have noted the salient prior work relevant to this paper and tried to point out where we are adding something new

With these concerns aside, there are questions about how the semantic analysis was conducted. How were the inter-relations between the semantic scores subjected to a principal component analysis? I can imagine how that was done, but providing the details would help. 

 These details are now provided with more elaborate explanations.

The more important analysis of the relation between the semantic scores and conspiracy beliefs is totally opaque. It is not at all clear what is meant by the sum of factor loadings in Table 6 or how those were related to ratings of conspiracy beliefs. Nor is there any clear interpretation of the second factor that is claimed to correlate with conspiracy thinking. 

Hopefully this is now easier to understand. The semantic analyses build on a widely published framework, now extensively cited and explained in the text. The techniques are explained and compared with the empirically obtained survey responses. In the most important table, number 6, we show that the semantic and empirical relationships are correlated .70 (p < .01) which hardly suggests a serendipitous result. The table should make it clear that the cognitive structures of the survey item texts alone show strongest relationships between cluster A scales and conspiration theories, and this pattern is detectable in the respondent data.

The authors seem to gloss over the finding that conscientiousness is related to conspiracy thinking. Why is that? Given the unequal reliabilities of the various measures used to assess correlates of conspiracy thinking, regression analysis is a poor guide to detecting true relations. I would go with the simple correlations more so than the regressions.

We have noted and discussed the conscientiousness finding. We have kept in the regressions recognizing that the problems highlighted by the reviewer are very common in this area.

The authors claim that the two measures of conspiracy mentality are only modestly correlated (.58) but given the reliabilities of the measures of .60 and .91 (as reported in the Discussion when it should be in the Methods), one would only expect a correlation of about .54. So, the correlation that was observed suggests that they two scales are virtually identical.

We have changed the “modestly” correlated to “strongly” correlated though we cannot accept that they are virtually identical.

Reviewer #2: Dear authors,

Thank you for your answers and comments. The manuscript is improved, but there are still some minor issues that should be addressed before the acceptance for publication:

Introduction

- Please add a short explanation of similarities and differences between 2 measures of CTs since you built your case based on these 2 measures.

See above…we have done this.

- Seems like the whole introduction is based on belief in CTs, and not conspiracy thinking. In the description of previous research please specify for each study which CTs measure is used

(specify the name of the measure, i.e., in Walter and Drochon study, etc.). This would contribute to a better understanding of the introduction of 2 measures of CT.

We have tried to do that. We have also tried to explain similarities and differences between Belief if CTs and conspiracy thinking, and further we have changed the title into “…conspiracy thinking and beliefs” to reflect this.

- Although the authors stated that they delete the term cryst. int., it is still present in the hypothesis: "We also used a short measure of crystalised intelligence to test the hypothesis that higher IQ test scores would be negatively correlated with belief in CTs (H2)." as well as in the whole text (e.g., in the discussion they used the term intelligence and not cryst. int.). 

Now done

Please specify that you expect negative relations between the general knowledge as the indicator of cryst. int. and CTs, or just - negative relations between the general knowledge and CTs.

The latter…now done

In addition, in this hypothesis, only beliefs in CTs are mentioned and not conspiracy thinking or CTs in general, as in other hypotheses.

OK…rectified to be consistent

Results

- There is no M and SD for CT2 in Table 1, please merge Tables 1 and 3. Please add alphas for all measures in Table 1 and report either sum or mean scores for used instruments (mean scores are better). Please delete descriptives for sex, it is odd. Add a note with variable explanation in all tables.

Done as requested

- I am not sure why in LSA only CT1 was entered, and not CT2. Please explain it at the beginning of the analysis. Are there some technical issues? I am not familiar with LSA and I could assume that readers would not be also familiar, thus the comment "(CT2 was not part of the analysis at all and the number is statistically significant even if N (the number of scales) is only 14)" should be explained more clearly. 

This passage is now changed. The reason CT2 is taken out is that long matrices of single items can get unwieldy in LSA, since the number of unique item pairs expands with every item added. We believe that the statistics and whole picture is easier to understand using only CT1, since the two are virtually interchangeable in effects, as per the comments of one of the reviewers.

- Seems like the rotation was not used in PCA, was it on purpose? Also, the statement "Here, the CT1 items have the strongest loadings, followed by the items from schizotypal scale" is not true, since the Depressive scale had loading -.216. 

To change the presentation of data, we have now shown a factor structure of the SCATI and CT1 based on a PCA with oblimin rotation. The ensuing picture is clearer. Negative loadings are tricky from an interpretive point of view in semantics and we have now tried to make the display of data more aligned with usual statistical representations.

- The following analysis presented in Table 6 is not clear to me and it is not clear why GK is in the table. The answer on your H9 could be get based on the pattern structure of the factors in Table 5 (with the use of rotation). If you have other rationale, please explain. Also sentence "If the CT1 factor loadings are removed from the table..." should be changed to "removed from the analysis..." 

 This is now all changed.

Discussion

- The discussion lack similarities and differences between CT1 and CT2. In addition, the statement "In this study we measured a number of individual variables which accounted for

22% of the variance in both analyses" should be replaced with - around 20%, since CT1 was explained by 20% and CT2 by 21%. 

Now changed thank you.

Other

- Check APA for use of capital letters (it should be personality disorder and conspiracy theories, but Dark Triad and not dark-triad or dark triad)

Done as requested

- Delete (Crystallised IQ) from the explanation of the General Knowledge Test.

Done as requested

- Delete M and SD from the description of the measures since they are already presented in Table 1.

Done as requested 

- Results for model testing remove from Table 2 (Model Adj. R2 = .20. F = 6.868, p < .001) and Table 4 (Model Adj. R2 = .21. F = 7.250, p < .001) and report in text as you did for next analysis.

Done as requested 

- Please specify on p.10 that you use hierarchical regression analysis.

Done as requested

We very much hope that you can accept this revision for the journal

---

## [Decision Letter · Decision Letter 2]

15 Aug 2022

Dispositional and ideological factor correlate of conspiracy thinking and beliefs

PONE-D-21-40074R2

Dear Dr. Arnulf,

We’re pleased to inform you that your manuscript has been judged scientifically suitable for publication and will be formally accepted for publication once it meets all outstanding technical requirements.

Kind regards,

Goran Knežević

Academic Editor

PLOS ONE

Additional Editor Comments (optional):

Reviewers' comments:

Reviewer's Responses to Questions

**Comments to the Author**

1. If the authors have adequately addressed your comments raised in a previous round of review and you feel that this manuscript is now acceptable for publication, you may indicate that here to bypass the “Comments to the Author” section, enter your conflict of interest statement in the “Confidential to Editor” section, and submit your "Accept" recommendation.

Reviewer #2: All comments have been addressed

2. Is the manuscript technically sound, and do the data support the conclusions?

Reviewer #2: Yes

3. Has the statistical analysis been performed appropriately and rigorously? 

Reviewer #2: Yes

4. Have the authors made all data underlying the findings in their manuscript fully available?

Reviewer #2: Yes

5. Is the manuscript presented in an intelligible fashion and written in standard English?

Reviewer #2: Yes

6. Review Comments to the Author

Reviewer #2: Dear authors,

Thank you for your answers and corrections that you made. All my suggestions were incorporated in the manuscript, thus it could be accepted.

7. PLOS authors have the option to publish the peer review history of their article (what does this mean?). If published, this will include your full peer review and any attached files.

Reviewer #2: **Yes: **Bojana M. Dinić

---

## [Editor Report · Acceptance letter]

28 Sep 2022

PONE-D-21-40074R2 

 Dispositional and ideological factor correlate of conspiracy thinking and beliefs 

Dear Dr. Arnulf:

I'm pleased to inform you that your manuscript has been deemed suitable for publication in PLOS ONE. Congratulations! Your manuscript is now with our production department. 

Kind regards, 

on behalf of

Prof Goran Knežević 

Academic Editor

PLOS ONE